# Problematic Internet Use and Resilience: A Systematic Review and Meta-Analysis

Sergio Hidalgo-Fuentes [1,2], Manuel Martí-Vilar [2,*] and Yolanda Ruiz-Ordoñez [3]

1. Departamento de Psicología y Salud, Universidad a Distancia de Madrid (UDIMA), Crta. De la Coruña Km. 38,500, vía de Servicio Número 15, Collado Villalba, 28400 Madrid, Spain
2. Departamento de Psicología Básica, Universitat de València, Avgda. Blasco Ibañez 21, 46010 Valencia, Spain
3. Departamento de Psicología Básica, Neuropsicología y Social, Universidad Católica de Valencia, 46100 Burjassot, Spain
* Correspondence: manuel.marti-vilar@uv.es

**Abstract:** Problematic Internet use has become a major problem worldwide due to its numerous negative correlates in the field of health, both mental and physical, and its increasing prevalence, making it necessary to study both its risk and protective factors. Several studies have found a negative relationship between resilience and problematic Internet use, although the results are inconsistent. This meta-analysis assesses the relationship between problematic Internet use and resilience, and analyses its possible moderating variables. A systematic search was conducted in PsycInfo, Web of Science and Scopus. A total of 93,859 subjects from 19 studies were included in the analyses. The results show that there is a statistically-significant negative relationship ($r = -0.27$ (95% CI [$-0.32, -0.22$])), without evidence of publication bias. This meta-analysis presents strong evidence of the relationship between the two variables. Limitations and practical implications are discussed.

**Keywords:** problematic Internet use; Internet addiction; resilience; meta-analysis





## 1. Introduction

Internet use has grown substantially over the last few decades, with the number of users increasing by 1331.9% between 2000 and 2021 [1], when a total of 4.66 billion users were counted, representing approximately 60% of the world's population [2]. The benefits associated with using the Internet, especially concerning information search and communication, have led people to rely more and more on this technology for their work, study, social interaction and access to various entertainment options [3]. However, excessive and uncontrolled use of this technology can lead to what has been termed problematic Internet use (PIU), which is defined as Internet use that causes psychological, social, educational and/or occupational difficulties in an individual's life [4]. Although the term Internet addiction, conceptualized as an impulse control disorder whereby the person loses control over their use of the Internet to the extent that they experience numerous negative consequences, as proposed by Young [5], is widely used in the scientific literature [6], a considerable number of authors recommend the use of PIU as more appropriate [7–9], since it is not recognised as an addictive disorder in either the DSM-5 [10] or ICD-11 [11].

PIU has been associated with numerous negative variables related to both mental and physical health, such as anxiety and depression [12], low self-esteem [13], poor sleep quality [14], alexithymia [15], risk of obesity [16], high impulsivity [17] and problematic alcohol consumption [18], among others, and the World Health Organization has declared PIU a major public health concern, emphasizing the need to intensify international research on this problem to generate the information required to develop policies and interventions to prevent and treat PIU [19]. A recent meta-analysis, which was conducted on a total sample of 2,123,762 people, has estimated the prevalence of PIU among the general population at 14.22% [20], having increased in recent years [21]. The high number of detrimental

variables associated with PIU, as well as its increasing prevalence, makes it necessary to emphasize the study of both potential risk factors and protective factors for PIU.

How to conceptualize resilience is a widely debated topic in the field of psychology [22]. Resilience can be defined as an individual's ability to maintain or regain psychological well-being in the face of a challenging situation [23]; it is a dynamic process that encompasses positive adaptation in the face of significant adversity, which would include feedback, learning and making changes to remain positive and recover from frustration caused by stressful events [24]. Resilience is an important factor in personal well-being, being negatively correlated to negative indicators of mental health, such as depression and anxiety, and positively correlated to positive indicators of mental health, such as life satisfaction and positive affect [25]. Several studies have examined the role of resilience in various types of addictive behaviors, and have found that resilience serves as a protective factor against addiction to gambling [26], alcohol [27,28], drugs of abuse [29,30], and video games [31,32]. Likewise, the relationship between resilience and PIU has also been evaluated, and has found negative relationship between both variables [33–35]. However, to date there has been no meta-analysis specifically focused on the relationship between PIU and resilience that synthesizes the results found. The aim of this paper is therefore to synthesize the evidence from those studies that have examined the association between PIU and resilience by answering the following research questions: (1) what is the strength of the association between PIU and resilience?; and (2) is the association between PIU and resilience moderated by the methodological and socio-demographic variables of the studies analyzed?

## 2. Materials and Methods

### 2.1. Systematic Search

This meta-analysis (registered in PROSPERO database #CRD42022382337) was conducted according to the criteria of the PRISMA statement [36] (Appendix A, Table A1). A systematic search was conducted during November 2022 in three databases (PsycINFO, Scopus and Web of Science) using the terms (resilience OR resiliency OR resilient) AND (internet addiction OR problematic internet use OR internet abuse OR internet overuse OR internet dependence). Searches were restricted to papers published in English or Spanish. Moreover, the references of the selected articles were manually checked for other relevant studies that were not retrieved during the electronic search. The systematic reviews software Covidence (http://www.covidence.org accessed on 14 November 2022) was used to manage the study selection process.

### 2.2. Inclusion Criteria

The retrieved studies were selected based on the following inclusion criteria: (1) original empirical and quantitative cross-sectional or longitudinal studies; (2) published in peer-reviewed scientific journals; (3) published in English or Spanish; (4) include assessments of PIU and resilience; (5) present Pearson's correlation coefficient between PIU and resilience or the statistical data necessary to calculate it: (6) present the sample size; and (7) the full text was available. In case of studies with partially duplicated samples, the study with the largest sample size was selected.

### 2.3. Methodological Quality of Included Studies

Conducting a meta-analysis without taking into consideration the methodological quality of the included studies may lead to biased results. Therefore, an assessment of the methodological quality of the studies analyzed in a meta-analysis is essential to be able to draw reliable conclusions. The risk of individual bias of the studies included in the meta-analysis was assessed using the short version of the Newcastle-Ottawa scale developed by Deng et al. [37]. The scale consists of a total of five items: (1) representativeness of the sample (inclusion of the entire population or random sampling); (2) sample size justified by methods such as power analysis; (3) response rate greater than 80%; (4) valid PIU and

resilience assessment tests; and (5) appropriate and correctly described statistical analyses. Each item is scored as one point if it meets the criterion and zero points if it does not meet the criterion or the information is not available. The total score ranges from zero to five points, with studies scoring three or more points being considered at low risk of individual bias and those scoring less than three points being considered at high risk of individual bias. Assessments were performed by two reviewers working independently. Discrepancies were resolved by consensus.

### 2.4. Data Coding

A recording sheet was prepared to code the following information for the studies included: author(s), year of publication, country in which the study was conducted, continent, sample size, mean age of participants, gender (coded as the percentage of males in the sample), test used to assess PIU, test used to assess resilience, risk of individual bias and Pearson's correlation between PIU and resilience. Data coding was performed by two reviewers working independently. The reviewers matched their data after extraction and revisited papers in case of disagreements. In the event of missing data, we contacted the authors of the study to request the necessary information; where we received no response or the authors refused to provide it, the information is listed as missing. To meet the independence assumption, in the case of longitudinal studies only the first correlation between PIU and resilience was coded.

### 2.5. Data Analysis

Most of the studies had Pearson correlations. For those studies with $\chi^2$, this result was converted to Pearson correlations using the formula $r = \sqrt{(\chi^2/n)}$. Subsequently, to normalize their distributions, all Pearson correlations were converted to Fisher's Z-scores using the formula $Z = 0.5 \times \ln[(1 + r)/(1 - r)]$. All analyses were performed with Z-scores, although the overall effect size and its confidence interval were transformed back to Pearson correlations for better interpretation following the recommendation of Borenstein et al. [38].

Due to the variability observed in the selected studies in terms of the countries in which they were conducted, the number of subjects and tests used, a random-effects meta-analysis with the restricted maximum likelihood method was chosen. Random-effects models generally produce more precise estimates and allow for greater generalizability of results [39–41]. The existence of statistically significant heterogeneity among the effect sizes of the analyzed studies was examined using Cochran's $Q$ test, while the degree of true heterogeneity not explained by random sampling error was assessed using the $I^2$ statistic. $I^2$ values of 25%, 50% and 75% are interpreted respectively as low, moderate and high heterogeneity [42].

The validity of a meta-analysis may be challenged by the presence of publication bias, a phenomenon whereby studies with statistically significant results or high effect sizes are more likely to be published [43]. Publication bias is a particularly important problem when conducting meta-analyses, since it can lead to overestimated effect sizes. In this study, and as recommended by Botella and Sánchez-Meca [44], the risk of publication bias was assessed by several methods: visual inspection of the funnel plot, Egger's regression test [45], Begg and Mazumdar's rank correlation test [46], and calculating the safety number according to Rosenthal's method. In the absence of publication bias, the funnel plot will be symmetrical around the average effect size, while Egger's test and Begg and Mazumdar's test will show non-significant results. Rosenthal's method makes it possible to estimate missing studies to calculate how many studies would be required for the estimated effect size to be non-significant.

A jacknife sensitivity analysis was performed, estimating the pooled effect size while eliminating each study alternatively, to assess the individual influence on the overall effect size of each of the studies included in the meta-analysis.

We examined the possible moderating role of the following variables: sex and age of participants, measures for assessing PIU and resilience, the continent in which the studies

were conducts, individual risk of bias and year of publication. For continuous variables, meta-regression analyses were conducted, while for categorical variables, subgroup analyses were conducted. For subgroup analysis, and as recommended by Fu et al. [47], each subgroup should be composed of a minimum of four studies. When this was not possible due to fewer studies having been performed, the remaining studies were grouped into the subgroup others and included in the analyses under this heading if they comprised at least four studies. The percentage of variance explained by the moderators was assessed using the $R^2$ index.

Analyses were performed in R Studio using the metafor statistical package [48].

## 3. Results

As can be seen in Figure 1, the search and selection process ended with the inclusion of 19 studies that met the inclusion criteria. The selected articles were published between 2015 and 2022 (see Table 1). Eight of the studies were conducted in China, four in South Korea, two in the United States and Turkey, and one each in Australia, Hungary and Iran. The combined sample was 93,859 subjects, with the sample sizes of the various studies ranging from 96 to 58,756 participants.

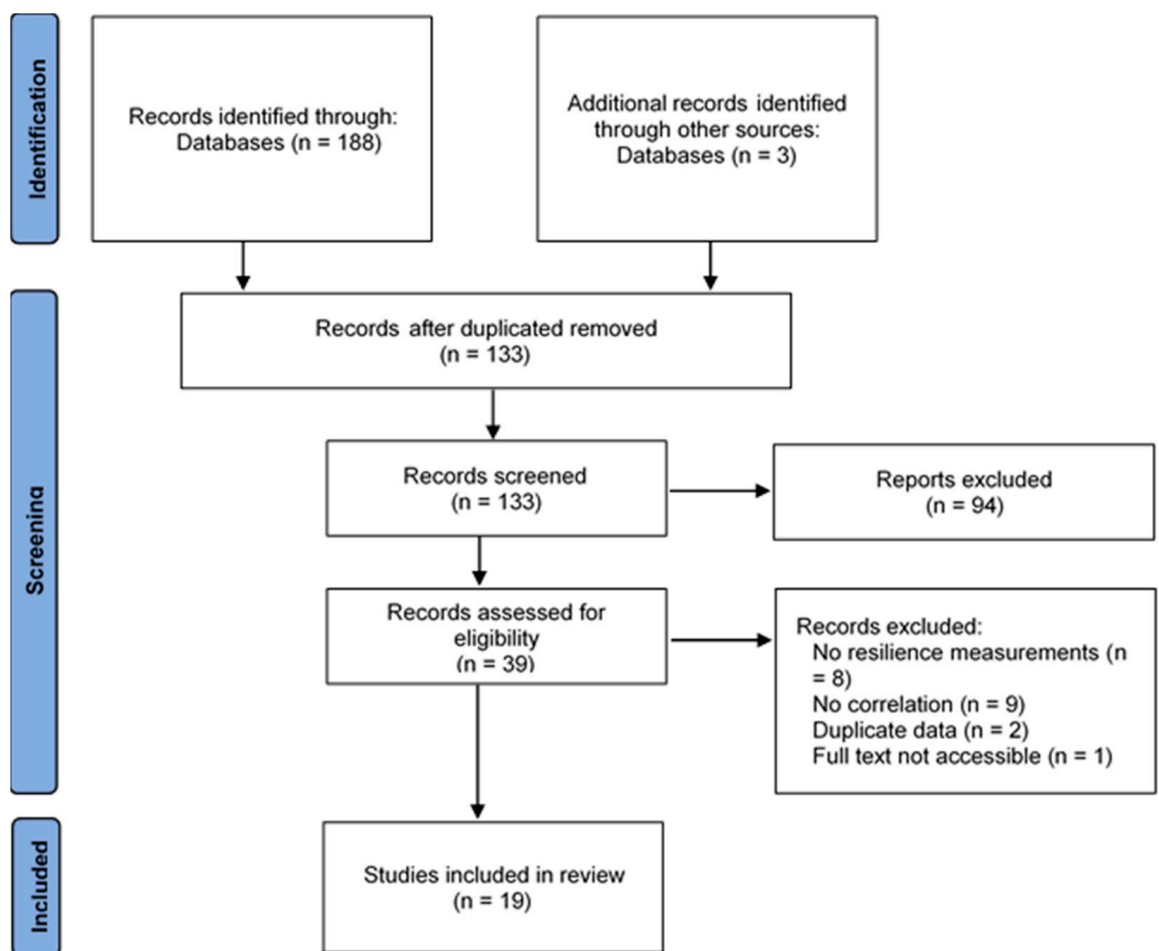

**Figure 1.** PRISMA diagram of the search and selection process.

**Table 1.** Summary of studies included in the meta-analysis.

| Study | Country | Continent | Sample | Age | Sex (% Men) | PIU Test | Resilience Test | Risk of Bias | r |
|---|---|---|---|---|---|---|---|---|---|
| Cao et al., 2020 [49] | China | Asia | 1218 | 11.8 | 55.25 | YDQ | CD-RISC 10 | Under | −0.214 |
| Choi et al., 2015 [50] | South Korea | Asia | 448 | 20.89 | 39.7 | IAT | CD-RISC | Under | −0.12 |
| Cui & Chi, 2021 [51] | China | Asia | 2544 | 16.49 | 42.7 | YDQ | CD-RISC 10 | Under | −0.267 |
| Dinc & Topcu, 2021 [33] | Australia | Oceania | 220 | 14.16 | 44.5 | CIUS | CYRM-28 | High | −0.29 |
| Dong & Li, 2020 [52] | China | Asia | 1362 | | 53.9 | IAII | CD-RISC 10 | Under | −0.25 |
| Hsieh et al., 2021 [53] | China | Asia | 6233 | | 51 | CIAS | CD-RISC 10 | Under | −0.17 |
| Jin et al., 2019 [54] | USA | America | 326 | 23.4 | 20.6 | IAT | BRS | Under | −0.121 |
| Kiss et al., 2020 [55] | Hungary | Europe | 249 | 22.5 | 37.8 | PIU-Q | CD-RISC 10 | High | −0.274 |
| Lee et al., 2022 [56] | South Korea | Asia | 866 | | 70.8 | IAPS | CD-RISC | High | −0.39 |
| Mak et al., 2018 [57] | South Korea | Asia | 837 | 22.13 | 43.13 | IAT | CD-RISC | High | −0.4 |
| Nam et al., 2018 [58] | South Korea | Asia | 519 | | 51.64 | IAT | CD-RISC | High | −0.122 |
| Öztürk & Kundakçı, 2021 [34] | Turkey | Europe | 1028 | 20.17 | 39.7 | IAT | BRS | Under | −0.498 |
| Peng et al., 2021 [59] | China | Asia | 16,130 | 15.22 | 51.9 | IAT | RSCA | Under | −0.252 |
| Robertson et al., 2018 [35] | USA | America | 240 | 25.05 | 65 | IAT | CD-RISC | High | −0.36 |
| Saeed, 2020 [60] | China | Asia | 436 | 23.81 | | IAT | BRS | High | −0.15 |
| Salek-Ebrahimi et al., 2019 [61] | Iran | Asia | 96 | 19.73 | 21.1 | IAT | CD-RISC | Under | −0.222 |
| Yilmaz et al., 2022 [62] | Turkey | Europe | 1123 | 46.7 | 58 | YIAT-SF | BRS | Under | −0.346 |
| Zhang & Li, 2022 [63] | China | Asia | 1228 | | | YDQ | PPQ | High | −0.38 |
| Zhou et al., 2017 [64] | China | Asia | 58,756 | 10.83 | 54.5 | YDQ | RRS | High | −0.218 |

YDQ: Young's Diagnostic Questionnaire for Internet Addiction; IAT: Young's Internet Addiction Test; CIUS: Compulsive Internet Use Scale; IAII: Internet Addiction Impairment Index; CIAS: Chen Internet Addiction Scale; PIU-Q: Problematic Internet Use Questionnaire; IAPS: Korean Internet Addiction Proneness Scale for Youth; YIAT-SF: Young's Internet Addiction Test-Short Form; CD-RISC 10: Connor-Davidson Resilience Scale Short Form; CD-RISC: Connor-Davidson Resilience Scale; CYRM-28: Child and Youth Resilience Measure; BRS: Brief Resilience Scale; RSCA: Resilience Scale for Chinese Adolescents; PPQ: PsyCap Questionnaire; RRS: Revised Resilience Scale.

The estimated overall effect size for the correlation between PIU and resilience was $Z_r = -0.28$ (95% CI [−0.33, −0.22]), which transformed back to Pearson's correlation gives a result of $r = -0.27$ (95% CI [−0.32, −0.22]), and which, following the interpretation criteria proposed by Cohen [65], can be classified as a moderate intensity correlation. The forest plot of the effect sizes and 95% confidence intervals of the 19 studies are shown in Figure 2. As can be seen in the figure, the effect sizes of the studies ranged from $Z_r = -0.12$ to $Z_r = -0.55$. The Cochran's $Q$ test result was 281.4128, $p < 0.0001$, hence the homogeneity hypothesis is rejected, while the $I^2$ value reached a value of 97.46%, which is considered high according to Higgins and Thompson's criteria [42].

Although the funnel plot is not fully symmetrical (see Figure 3), both the Egger regression test ($z = 0.2996$, $p = 0.76$) and the Begg and Mazumdar rank correlation test ($\tau = -0.0292$, $p = 0.89$) show non-significant results, thus ruling out the presence of publication bias. Likewise, the calculation of the number of safety according to Rosenthal's method yielded a value of n = 18,877 ($p < 0.001$), making 18,877 unpublished studies with an effect size equal to zero necessary to make the $p$-value non-significant, exceeding the critical value which, for this meta-analysis, is set at 105 studies, according to the formula (5 × k) + 10, and k being the number of studies included in the meta-analysis [44].

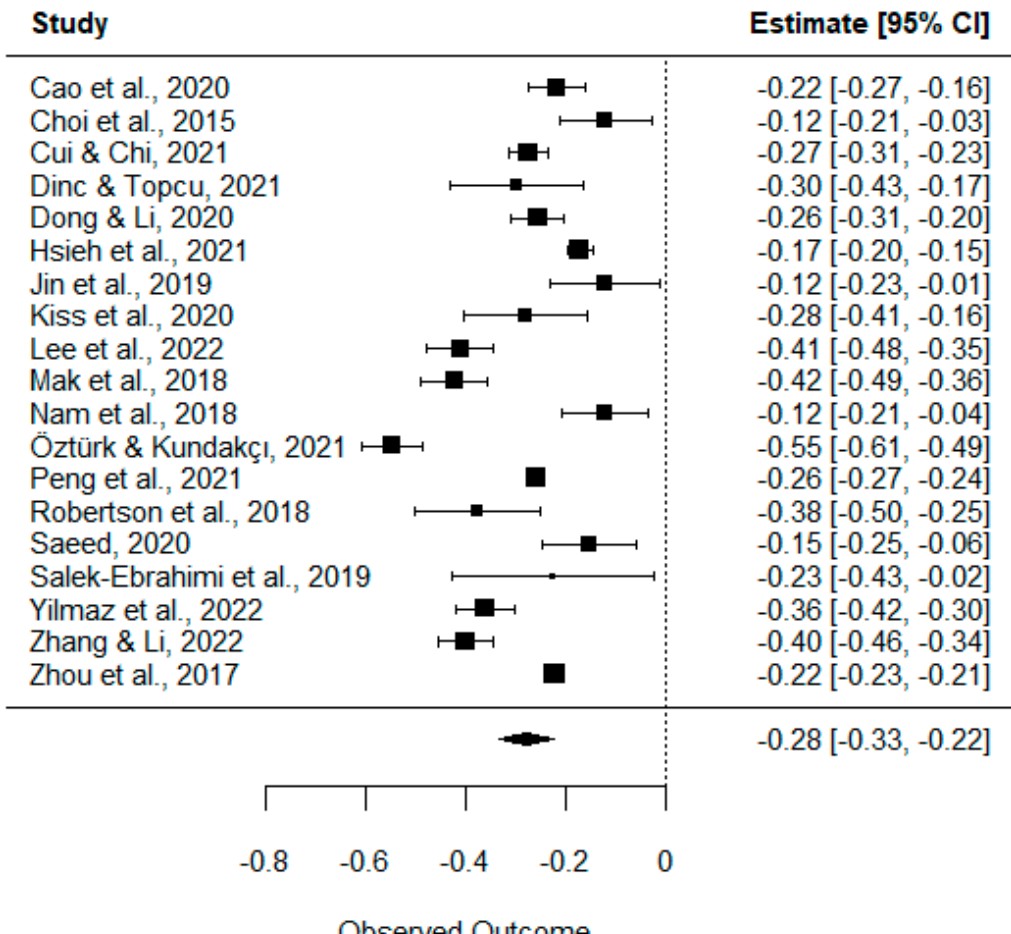

**Figure 2.** Effect size for the relationship between PIU and resilience [33–35,49–64].

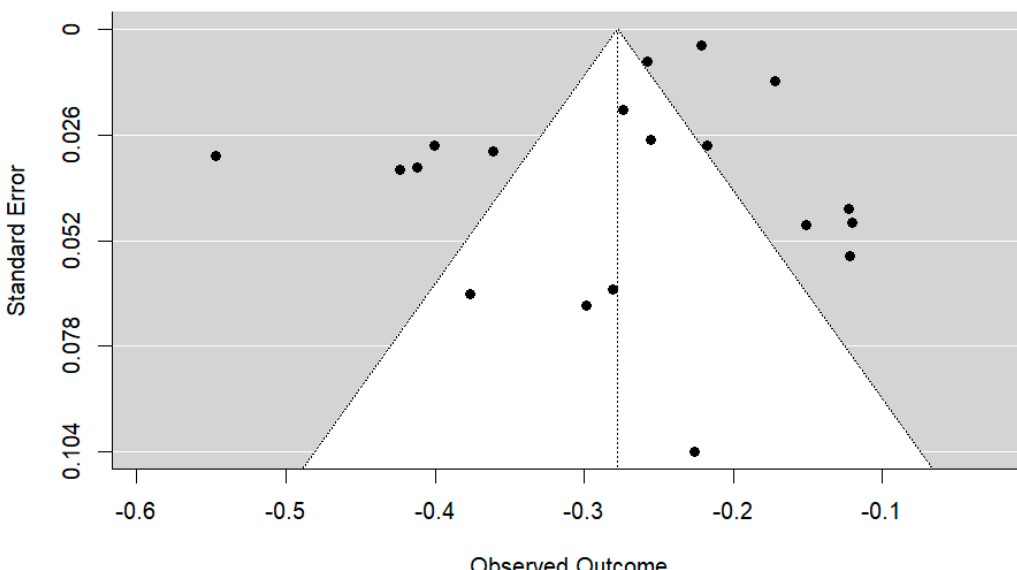

**Figure 3.** Funnel plot for assessing publication bias.

The sensitivity analysis, performed using the jackknife method, did not show excessive individual influence of any of the studies on the estimated overall effect size, with the effect size ranging from $Z_r = -0.26$ to $Z_r = -0.29$ when alternately omitting each of the studies.

Meta-regression analyses were conducted to examine the possible moderating effect of year of publication, mean age of participants and percentage of males in the sample on the correlation between PIU and resilience. Both mean age ($b = -0.0028$, $p = 0.72$) and percentage of males among participants ($b = -0.0027$, $p = 0.26$) did not show up as moderating variables, while year of publication ($b = -0.0283$, $p = 0.04$) does moderate the relationship between the two variables, with a total explained variability of 16.55%, with more recent studies showing lower correlations.

For categorical variables, subgroup analyses were performed (see Table 2), and no moderating effect was found for any of the variables analyzed.

**Table 2.** Relationship between PIU and resilience: moderation analysis for categorical variables.

| | $Z_r$ | 95% CI | $p$ | $p$ Subgroup |
|---|---|---|---|---|
| Risk of individual bias | | | | 0.48 |
| High | −0.30 | −0.38, −0.22 | <0.001 | |
| Under | −0.26 | −0.33, −0.18 | <0.001 | |
| Continent | | | | 0.15 |
| Asia | −0.25 | −0.32, −0.19 | <0.001 | |
| Other | −0.34 | −0.44, −0.24 | <0.001 | |
| PIU test | | | | 0.90 |
| IAT | −0.27 | −0.35, −0.18 | <0.001 | |
| YDQ | −0.28 | −0.40, −0.16 | <0.001 | |
| Other | −0.30 | −0.40, −0.19 | <0.001 | |
| Resilience test | | | | 0.87 |
| BRS | −0.30 | −0.43, −0.18 | <0.001 | |
| CD-RISC | −0.28 | −0.39, −0.18 | <0.001 | |
| CD-RISC 10 | −0.24 | −0.35, −0.13 | <0.001 | |
| Other | −0.29 | −0.42, −0.17 | <0.001 | |

## 4. Discussion

The first aim of this paper was to estimate the magnitude of the association between PIU and resilience. Additionally, we examined the possible moderating role of gender and age of participants, the continent on which the studies were conducted, the tests used to assess both PIU and resilience, the year of publication of the studies, and the risk of individual bias.

The systematic search identified a total of studies that met the inclusion criteria with a total sample of 93,859 subjects. The results of the meta-analyses showed a statistically significant negative correlation of moderate intensity ($r = -0.27$) between the two variables, whereby those who showed higher levels of resilience had lower levels of PIU. Sensitivity analysis reveals that this result is consistent, with none of the studies having an excessive influence on the overall effect size. Furthermore, the various tests performed to assess the risk of publication bias ruled out the presence of bias. Despite the high degree of heterogeneity found, only the year of publication proved to be a moderating variable in the correlation between PIU and resilience, explaining 16.55% of the observed heterogeneity.

The result found has important implications for the prevention of PIU, a phenomenon with significant negative repercussions on mental and physical health, as well as significant associated economic costs [66]. Resilience may function as a protective factor for PIU by mitigating the negative impact of adverse situations or environments, causing individuals to suffer lower levels of depression or anxiety [67], two variables that have been consistently linked in the scientific literature to PIU [12,68–70]. Additionally, in theoretical terms, the negative association found between resilience and PIU could be explained in relation to the I-PACE model, which explains the onset and development of PIU by the interaction of personal, affective, cognitive and executive variables [6]. This theoretical model holds that stress is an important factor operating on addictive behaviors and that excessive and uncontrolled use of the Internet can sometimes be a coping style that attempts to cope with

this stress. Resilience also improves people's ability to cope with stressful situations, which are also a risk factor for PIU [71], as individuals with high levels of stress often use the Internet as a maladaptive coping strategy because, although it does not offer long-term improvement, Internet use can serve as a temporary relief from stressful symptoms. Thus, from this perspective, resilience, which is taken to be the ability to cope with adverse and stressful situations, may lead to a lesser need to use the Internet to reduce stress levels, since resilience itself will act as a protective factor. Thus, people with higher levels of resilience have and make use of adaptive coping strategies in stressful situations, which may prevent them from engaging in compulsive behaviors such as PIU. Therefore, the results obtained, together with the fact that resilience can be increased through appropriate programs [72], allow us to state that interventions aimed at increasing resilience can be an effective method of reducing the risk of PIU. Besides preventing the onset of PIU, resilience has also shown benefits when IPU has already developed, serving as a protective factor against the negative psychological effects of PIU [73].

Among the possible moderating variables of the relationship between PIU and resilience examined, the only statistically significant moderator was the studies' year of publication, with more recent articles showing a smaller effect size among the variables studied. One possible explanation for this is that the more recent studies, conducted during the pandemic when many countries were in lockdown, show a lower relationship between PIU and resilience since individuals during this period suffered greater stress that could not be compensated for by their resilience levels, leading to excessive internet use to reduce this stress. By contrast, participants' gender and age, as well as the geographical area in which the studies were conducted, are not statistically significant moderators of the relationship between PIU and resilience. The fact that there is little heterogeneity regarding these variables, especially age and geographic area, in the included studies could be influencing this result.

The results of this meta-analysis should be interpreted with caution due to certain limitations. Firstly, the number of studies that met the inclusion criteria is limited, so it would be advisable for future systematic reviews or meta-analyses to extend the search to other databases. Secondly, only studies published in Spanish or English were included, which could be considered a selection bias, despite English being the most widely used language in the scientific literature. Thirdly, only one of the possible moderating variables was found to have a significant effect and it could not explain a significant percentage of the heterogeneity found. It would therefore be important for future meta-analyses to examine the role of new potential moderators of the correlation between PIU and resilience, such as the population in which the studies were carried out or the scores obtained. Fourth, given the cross-sectional design of most of the included studies, it is not possible to establish causal relationships between the variables analyzed or to examine their evolution over time, hence it would be desirable to conduct further longitudinal or experimental design research in the future to examine these matters. Finally, most of the studies were conducted in Asian countries and with adolescent and young participants, with very limited research in other geographical areas and with subjects in other age groups.

## 5. Conclusions

PIU has become a growing problem in recent years, especially among adolescents and young people, being associated with many harmful variables, mainly psychological, hence studying its risk and protective factors to help to prevent and treat it should be a priority, bearing in mind both its negative effects and the number of people who suffer from this problem. This meta-analysis has synthesized the results on PIU and resilience. The results of this review, despite its limitations, indicate the existence of a significant negative relationship of moderate intensity between both variables that does not appear to depend on age, gender, geographical area or the tests used. This result has implications that go beyond the theoretical field by supporting the fact that working on people's resilience can reduce the risk of PIU. Moreover, increasing resilience levels through appropriate training

programs would have beneficial effects beyond reducing the risk of IPU, since resilience has also been shown to be a protective factor against other addictive behaviors such as alcohol consumption [27], gambling [26], drug abuse [29] and Internet gaming disorder [32]. Likewise, increasing resilience would also have a positive impact on other variables not directly related to problematic use of new technologies or addictions, improving both physical and mental health [72].

**Author Contributions:** Conceptualization, S.H.-F. and M.M.-V.; methodology, S.H.-F., M.M.-V. and Y.R.-O.; software, S.H.-F.; validation, S.H.-F., M.M.-V. and Y.R.-O.; formal analysis, S.H.-F.; investigation, S.H.-F., M.M.-V. and Y.R.-O.; resources, M.M.-V. and Y.R.-O.; data curation, S.H.-F.; writing—original draft preparation, S.H.-F.; writing—review and editing, M.M.-V. and Y.R.-O.; visualization, M.M.-V. and Y.R.-O.; project administration, M.M.-V. and Y.R.-O. All authors have read and agreed to the published version of the manuscript.

**Funding:** This research received no external funding.

**Institutional Review Board Statement:** Not applicable.

**Informed Consent Statement:** Not applicable.

**Data Availability Statement:** Not applicable.

**Conflicts of Interest:** The authors declare no conflict of interest.

## Appendix A

**Table A1.** Search Strings; PRISMA Checklist.

| Section/Topic | # | Checklist Item | Reported on Page # |
|---|---|---|---|
| **TITLE** | | | |
| Title | 1 | Identify the report as a systematic review, meta-analysis, or both. | 1 |
| **ABSTRACT** | | | |
| Structured summary | 2 | Provide a structured summary including, as applicable: background; objectives; data sources; study eligibility criteria, participants, and interventions; study appraisal and synthesis methods; results; limitations; conclusions and implications of key findings; systematic review registration number. | 1 |
| **INTRODUCTION** | | | |
| Rationale | 3 | Describe the rationale for the review in the context of what is already known. | 1–2 |
| Objectives | 4 | Provide an explicit statement of questions being addressed with reference to participants, interventions, comparisons, outcomes, and study design (PICOS). | 2 |
| **METHODS** | | | |
| Protocol and registration | 5 | Indicate if a review protocol exists, if and where it can be accessed (e.g., Web address), and, if available, provide registration information including registration number. | 2 |
| Eligibility criteria | 6 | Specify study characteristics (e.g., PICOS, length of follow-up) and report characteristics (e.g., years considered, language, publication status) used as criteria for eligibility, giving rationale. | 2 |
| Information sources | 7 | Describe all information sources (e.g., databases with dates of coverage, contact with study authors to identify additional studies) in the search and date last searched. | 2 |

**Table A1.** *Cont.*

| Section/Topic | # | Checklist Item | Reported on Page # |
|---|---|---|---|
| Search | 8 | Present full electronic search strategy for at least one database, including any limits used, such that it could be repeated. | Table A1 |
| Study selection | 9 | State the process for selecting studies (i.e., screening, eligibility, included in systematic review, and, if applicable, included in the meta-analysis). | 2 |
| Data collection process | 10 | Describe method of data extraction from reports (e.g., piloted forms, independently, in duplicate) and any processes for obtaining and confirming data from investigators. | 2–3 |
| Data items | 11 | List and define all variables for which data were sought (e.g., PICOS, funding sources) and any assumptions and simplifications made. | 2–3 |
| Risk of bias in individual studies | 12 | Describe methods used for assessing risk of bias of individual studies (including specification of whether this was done at the study or outcome level), and how this information is to be used in any data synthesis. | 2 |
| Summary measures | 13 | State the principal summary measures (e.g., risk ratio, difference in means). | 3 |
| Synthesis of results | 14 | Describe the methods of handling data and combining results of studies, if done, including measures of consistency (e.g., $I^2$) for each meta-analysis. | 3 |
| Risk of bias across studies | 15 | Specify any assessment of risk of bias that may affect the cumulative evidence (e.g., publication bias, selective reporting within studies). | 3 |
| Additional analyses | 16 | Describe methods of additional analyses (e.g., sensitivity or subgroup analyses, meta-regression), if done, indicating which were pre-specified. | 3 |
| **RESULTS** | | | |
| Study selection | 17 | Give numbers of studies screened, assessed for eligibility, and included in the review, with reasons for exclusions at each stage, ideally with a flow diagram. | 3–4 |
| Study characteristics | 18 | For each study, present characteristics for which data were extracted (e.g., study size, PICOS, follow-up period) and provide the citations. | 4–5 |
| Risk of bias within studies | 19 | Present data on risk of bias of each study and, if available, any outcome level assessment (see item 12). | 4–5 |
| Results of individual studies | 20 | For all outcomes considered (benefits or harms), present, for each study: (a) simple summary data for each intervention group (b) effect estimates and confidence intervals, ideally with a forest plot. | 5 |
| Synthesis of results | 21 | Present results of each meta-analysis done, including confidence intervals and measures of consistency. | 5–6 |
| Risk of bias across studies | 22 | Present results of any assessment of risk of bias across studies (see Item 15). | 5–6 |
| Additional analysis | 23 | Give results of additional analyses, if done (e.g., sensitivity or subgroup analyses, meta-regression [see Item 16]). | 6–7 |

**Table A1.** *Cont.*

| Section/Topic | # | Checklist Item | Reported on Page # |
|---|---|---|---|
| **DISCUSSION** | | | |
| Summary of evidence | 24 | Summarize the main findings including the strength of evidence for each main outcome; consider their relevance to key groups (e.g., healthcare providers, users, and policy makers). | 7 |
| Limitations | 25 | Discuss limitations at study and outcome level (e.g., risk of bias), and at review-level (e.g., incomplete retrieval of identified research, reporting bias). | 7 |
| Conclusions | 26 | Provide a general interpretation of the results in the context of other evidence, and implications for future research. | 7–8 |
| **FUNDING** | | | |
| Funding | 27 | Describe sources of funding for the systematic review and other support (e.g., supply of data); role of funders for the systematic review. | N/A |

From: [36]. For more information, visit: www.prisma-statement.org (accessed on 17 December 2022).

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
