# Peer review of "Problematic Internet Use and Resilience: A Systematic Review and Meta-Analysis"

_nursrep, doi:10.3390/nursrep13010032_

Round 1

Reviewer 1 Report

Congratulations for the authors, Very important results independently of the studies included.

Well written paper, very good description of the methodology.

The discussion and the conclusion should be extended.

The discussion should be more focused in the results of the meta analyses. There is not a strong description from the authors in relation to the studies involves/limitations/quality of the studies. 

Well done!

Author Response

Reviewer 1: We would like to thank reviewer 1 for his positive assessment and his comments. Following his suggestions, the discussion and conclusions have been expanded, with emphasis on the results found.

Additionally, in theoretical terms, the negative association found between resilience and PIU could be explained in relation to the I-PACE model, which explains the onset and development of PIU by the interaction of personal, affective, cognitive and executive variables [6]. This theoretical model holds that stress is an important factor operating on addictive behaviours and that excessive and uncontrolled use of the Internet can sometimes be a coping style that attempts to  cope with  this stress.

Thus, from this perspective, resilience, which is taken to be the ability to cope with adverse and stressful situations, may lead to a lesser need to use the Internet to reduce stress levels, since resilience itself will act as a protective factor. Thus, people with higher levels of resilience have and make use of adaptive coping strategies in stressful situations, which may prevent them from engaging in compulsive behaviours such as PIU.

Besides preventing the onset of PIU, resilience has also shown benefits when IPU has already developed, serving as a protective factor against the negative psychological effects of PIU [73].

Among the possible moderating variables of the relationship between PIU and resilience examined, the only statistically significant moderator was the studies’ year of publication, with more recent articles showing a smaller effect size among the variables studied. One possible explanation for this is that the more recent studies, conducted during the pandemic when many countries were in lockdown show a lower relationship between PIU and resilience since individuals during this period suffered greater stress that could not be compensated for by their resilience levels, leading to excessive internet use to reduce this stress. By contrast, participants’ gender and age, as well as the geographical area in which the studies were conducted, are not statistically significant moderators of the relationship between PIU and resilience. The fact that there is little heterogeneity regarding these variables, especially age and geographic area, in the included studies could be influencing this result.

Reviewer 2 Report

Dear authors, thank you for your manuscript. It is a very important issue to make research.

Considering that the article in question is a meta-analysis and seems to me to be well structured from a methodological point of view, as well as in the results presented, the methodological process of this research I think the manuscript is ok. 

Just review the text in line 141 (the text is bold and should not be). 

Thank you. 

Author Response

Reviewer 2: We thank the reviewer for his work and time spent supervising the manuscript. The paragraph on line 141 that was in bold has been corrected.

As can be seen in Figure 1, the search and selection process ended with the inclusion of 19 studies that met the inclusion criteria. The selected articles were published between 2015 and 2022 (see Table 1). Eight of the studies were conducted in China, four in South Korea, two in the United States and Turkey, and one each in Australia, Hungary and Iran. The combined sample was 93,859 subjects, with the sample sizes of the various studies ranging from 96 to 58,756 participants.

Reviewer 3 Report

The methodology, including statistical technique used in the study has been and has been adequately described. The conclusions drawn from Meta-Analysis are quite clear. 

Significant heterogeneity among studies suggests that there are covariate and/or moderating variables determining correlation between PIU and resilience.

Author Response

Reviewer 3: We thank the reviewer for his assessment and comments. As he rightly points out, the heterogeneity observed in the effect sizes of the various articles included points to the existence of moderating variables, which have included as a limitation of the article while also pointing out possible variables that could be examined in future studies.

Thirdly, only one of the possible moderating variables was found to have a significant effect and it could not explain a significant percentage of the heterogeneity found. It would therefore be important for future meta-analyses to examine the role of new potential moderators of the correlation between PIU and resilience, such as the population in which the studies were carried out or the scores obtained. Fourth, given the cross-sectional design of most of the included studies, it is not possible to establish causal relationships between the variables analysed or to examine their evolution over time, hence it would be desirable to conduct further longitudinal or experimental design research in the future to examine these matters.

Reviewer 4 Report

I would like to say thanks for the opportunity to review this article.

The article presented has a very interesting theme and important to the scientific community and community in general.

It presents the relation between resilience and Problematic internet use..

Overall, the article has a scientific and appropriate writing, including all the components of a good scientific research. The title and abstract are related with the content. The keywords are linked to the research and are included in Mesh terms. The article has chapters organized in a logic way.

Introduction allows the framing of the theme and the research itself. The main goal is appropriate.

Methodology is scientifically appropriate, and completely exposed.

Results are adequate and complete. Discussion is done according to the results of the study but does not explore theoretically the results in a profound way. Some results are not explored and explained with the scientific current evidence

Authors present limitations of the study and has appropriate conclusions.

References are recent in their majority (79% have less than 10 years) and are pertinent.

For this, we suggest the following corrections:

- the legends of figures and tables should be completed, allowing to know what is exposed in terms of results.

- Discussion should be enriched, specially with the explanation and discussion of some relevant results. The statistical results and methodology are analysed, but the discussion of the relations is poor and weakly supported. We would like to suggest some theme to discuss: is there an explanation why the year of publication has an impact on results? What could be interfering, considering the increase of internet use over the years? Is there an implication of PIU in resilience? Or the literature only justifies the opposite direction of implication? Why does age and gender not have an influence? You refer the country of the sample could have an implication of results. Is it possible to explore a little bit more what patterns of internet use in those countries could influence? What other variables are referred in literature like having an impact in this relation?

Thank you.

Author Response

Reviewer 4: We would like to thank the reviewer for his comments and suggestions. Following his suggestions, tables and figures have been re-titled to show more information. The discussion has been substantially expanded by adding some of the points made by the reviewer, such as a possible explanation of why year of publication is a statistically-significant moderator and why age and sex or geographic area are not. The impossibility of establishing causal relationships between the variables studied has also been explained, since most of the studies conducted on this subject are cross-sectional, and a recommendation has been included to continue studying the relationship between both variables using experimental and longitudinal studies. The conclusions have been extended.

Among the possible moderating variables of the relationship between PIU and resilience examined, the only statistically significant moderator was the studies’ year of publication, with more recent articles showing a smaller effect size among the variables studied. One possible explanation for this is that the more recent studies, conducted during the pandemic when many countries were in lockdown show a lower relationship between PIU and resilience since individuals during this period suffered greater stress that could not be compensated for by their resilience levels, leading to excessive internet use to reduce this stress. By contrast, participants’ gender and age, as well as the geographical area in which the studies were conducted, are not statistically significant moderators of the relationship between PIU and resilience. The fact that there is little heterogeneity regarding these variables, especially age and geographic area, in the included studies could be influencing this result.

One possible explanation for this is that the more recent studies, conducted during the pandemic when many countries were in lockdown show a lower relationship between PIU and resilience since individuals during this period suffered greater stress that could not be compensated for by their resilience levels, leading to excessive internet use to reduce this stress. By contrast, participants’ gender and age, as well as the geographical area in which the studies were conducted, are not statistically significant moderators of the relationship between PIU and resilience. The fact that there is little heterogeneity regarding these variables, especially age and geographic area, in the included studies could be influencing this result.

Fourth, given the cross-sectional design of most of the included studies, it is not possible to establish causal relationships between the variables analysed or to examine their evolution over time, hence it would be desirable to conduct further longitudinal or experimental design research in the future to examine these matters.